# Data-Driven Decision Making in Response to the COVID-19 Pandemic: A City of Cape Town Case Study

Elmarie Nel [1], Andrew MacLachlan [2,*], Ollie Ballinger [2], Hugh Cole [1] and Megan Cole [3]

1 Policy and Strategy Department, City of Cape Town, Cape Town 8000, South Africa
2 The Centre for Advanced Spatial Analysis (CASA), University College London, London WC1E 6BT, UK
3 Future Water Institute, University of Cape Town, Cape Town 7700, South Africa
* Correspondence: a.maclachlan@ucl.ac.uk

**Abstract:** In the event of a crisis, such as COVID-19, the decisions and subsequent actions taken by the local government are one of the primary sources of support to the local population. Yet the processes through which these decisions are reached and the data engineering advancements made for and during events are poorly reported. Understanding the capabilities and constraints in which city officials operate is essential for impactful academic research alongside global city comparison and discussion on best practices in reaching optimal and data-informed decisions. This is especially pertinent for the global South, where informality in housing and the economy presents further challenges to appropriate resource distribution in a crisis. Here, we present insights into the City of Cape Town's data-driven response and subsequent data engineering and analytical developments throughout the COVID-19 pandemic. This is based upon a review of internal documentation including a close-out report which summarised semi-structured interviews with staff involved in the data work stream. The paper reports on the deliverables produced during 2020 by the data work stream and outlines specific challenges the city faced and its data-informed response in the areas of (1) quantifying costs for COVID-19 initiatives, (2) dealing with a surge in fatalities, (3) guiding scarce public resources to respond to an evolving crisis, and (4) data sharing. We demonstrate the real-term value of incorporating data into the decision-making process and conclude by outlining key factors that cities and researchers must consider as a part of the usual business to effectively assist their populations during times of stress and crisis.

**Keywords:** data-driven policy; Cape Town; COVID-19 response; smart cities; urban spatial science; resilience

## 1. Introduction

The 2019 Coronavirus (COVID-19) pandemic presented a unique set of challenges for all levels of government, in many cases requiring the development of completely new work streams and operating procedures [1]. Although faced with the same virus, countries around the world encountered different constraints and opportunities in their responses to this unprecedented public health crisis [2]. In particular, countries with a high proportion of informality were confronted with additional difficulties related to the implementation of policies in hard-to-reach and vulnerable communities [3]. Cities were particularly hard hit by the pandemic, with impacts on revenues, rapidly rising resident needs and major challenges to maintaining basic/essential service delivery [4].

Cape Town, South Africa's second biggest metropolitan area with over 4.6 million residents in 2020 [5], has high levels of informality and communal water access, which hinder social distancing and good hygiene [6,7], high levels of HIV/AIDS and tuberculosis [8], which increase vulnerability to COVID-19, and an underfunded and overburdened public health care system [9]. However, it was able to respond quickly and avoid system failure, drawing on lessons learnt and data-driven tools developed during the 2018 Cape Town

drought crisis [10]. Just as with the drought, understanding the nature of the challenges faced by the City of Cape Town (CCT), as well as the strengths and weaknesses of its response to the pandemic, especially in utilising data-driven solutions, provides important lessons that can inform future data-driven responses to crises by city governments (ibid.).

COVID-19 was first reported to the World Health Organisation (WHO) on 31 December 2019 and has caused devastation across the world, with 6.4 million deaths recorded by July 2022 [11,12]. Mild symptoms include fever, dry cough and fatigue whilst in severe cases additional symptoms can include shortness of breath, high temperature and chest pain [11]. Of those who develop symptoms, the WHO estimates that 15% require oxygen and 5% need admission to intensive care and ventilators (ibid.). COVID-19 is known to spread through airborne particles with prevention tools (prior to widespread vaccination that commenced in December 2020) focusing on public health and social measures such as social distancing, hygiene, ventilation, personal protective equipment (PPE), self-isolation upon a confirmed case [13], reducing venue capacities and providing accessible distribution sites alongside financial assistance through centralised taxation. To limit the spread of COVID-19, governments implemented non-pharmaceutical interventions including lockdowns to entire countries or cities, restricting the movement and interaction of people to reduce the spread of the local epidemic and pressure on the healthcare system [13,14].

A critical element of the COVID-19 response effort involved tracking cases and modelling the potential spread through the population to ensure equitable distribution of protective resources and facilities. These resources were quickly exhausted, and a lack of medical supplies was seen in countries across the world, including Spain [15], the United Kingdom, the United States of America, Italy and India [16]. Médecins Sans Frontières (MSF), the WHO, the European Union, and the African Union subsequently established their own collective procurement process of PPE to remove inter-country competition [17].

Epidemiological models were relied on to guide decision-making. These typically consider characteristics of the virus and population (e.g., transmissivity and population behaviour) and the interaction of the virus with the population [18]. The R-value represented the average number of secondary infections from one infected person and was used to guide national governments on national or local lockdowns/restrictions and avoid overwhelming the healthcare system. Countries with informality needed to not only understand the spread of the virus but also the capacity and vulnerability of communities to adhere to mandated restrictions [6,19].

This paper will explore how the CCT's data-driven developments assisted in responding to these overwhelming challenges to optimise limited resources. Previous literature has discussed the impact of COVID-19 on the economy [20], cities in a post-COVID-19 world [21] and centralised government responses [22,23]. While some examples of data-driven responses to public health crises were published, such as New York City's response to Legionnaires disease in 2015 [24], often the internal data advances and work streams within metropolitan governments are poorly documented despite being at the forefront of city decision-making. In light of urbanisation and the need for urban resilience in developing countries, it is essential to understand the capabilities, constraints and processes in which government data analysts and policymakers operate to design and implement crisis responses. This paper describes one aspect of the City of Cape Town's COVID-19 response—the data work stream—in detail. It provides the methods used to document the case study and the main deliverables produced by the CCT. The Section 5 summarises the four major challenges and the CCT response: (1) quantifying costs for COVID-19 initiatives, (2) dealing with a surge in fatalities, (3) guiding scarce public resources to respond to an evolving crisis, and (4) data sharing and lessons learned.

The findings can be used to support similar departments and future collaborative academic research in driving forward data agendas and giving scientists a say in the future of cities [25]. In addition, they provide lessons about the use of data in shaping government responses to crises to inform future research on the role of data in enabling capable city governments.

## 2. South African and Cape Town Context

South Africa is an upper-middle-income country of 60 million people with very high levels of inequality, poverty and unemployment [26]. It has a relatively young population (44% under 25 years and only 6% over 65 years), an HIV prevalence rate of 13.9% and a life expectancy of 63 years old [5]. The South African economy was already in a weak position when it entered the pandemic after a decade of low growth due to longstanding structural challenges such as electricity shortages [27].

On 5 March 2020, South Africa recorded its first case of COVID-19 and recorded its first COVID-19-related death three weeks later [5]. From 12 March 2020, South Africa started to health screen international travelers arriving from Asia. This was swiftly escalated on 15 March in a Presidential announcement that discouraged all non-essential travel, limited gatherings to 100 people, and declaration of a national state of disaster (Figure 1). A full lockdown followed on 26 March, with all South Africans (except essential workers) required to stay at home except to purchase essential groceries. The national government announced a Temporary Employer–Employee Relief Scheme in March 2020 [28] and a special COVID-19 Social Relief of Distress grant in May 2020 [29]. On 1 May 2020, the South African government released a five-level alert plan to manage the gradual easing of the hard lockdown, ranging from a complete lockdown (level 5), to select industries operating (level 4), further retail opening (level 3), social distancing requirements (level 2) and most normal activity resuming (level 1) [30]. The alert level dropped from level 4 in May to level 3 in June, level 2 in August 2020 and level 1 in September (Figure 1). Many of the COVID-19 response measures resulted in significant reductions in wage income and even food security, particularly for low-skilled workers [31].

Amid rising cases at the start of 2021, level 3 was reintroduced in both January and May, before returning to level 1 in September (Figure 1). South Africa was the first country to detect the Omicron (BA.1) variant of COVID-19 in November 2021 but despite higher transmissibility, remained at level 1 given the previous extensive restrictions and lower hospitalisation rates compared to prior waves [32,33]. Whilst having a lower case load, the case fatality rate in South Africa has been decreasing less rapidly over time compared to many countries in the global North [34]. The national state of disaster was lifted on 5 April 2022 [13].

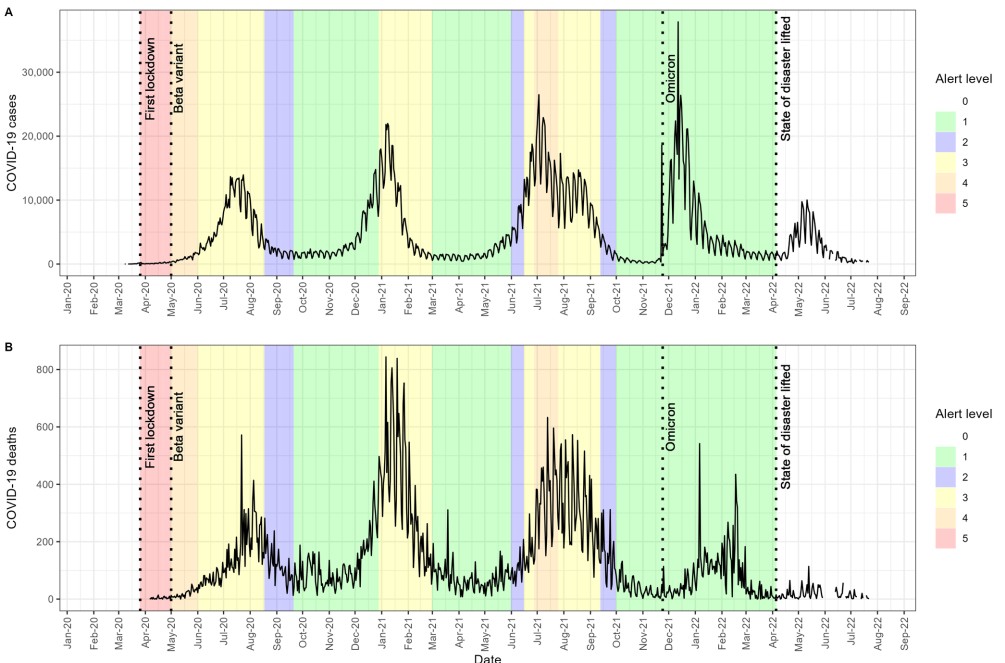

**Figure 1.** COVID-19 reported cases (**A**) and deaths (**B**) in relation to the South African alert plan and contextual information on COVID-19 variants. Data source: COVID-19 Data Hub [35].

The South African Constitution sets out individual responsibilities across national, provincial and local (municipal) governments. In the context of COVID-19, the national government oversaw the strategic direction in combatting COVID-19, whereas the provincial government oversaw hospital services and primary health care. The local government, such as in the CCT, is the sphere of government closest to the people, and delivers services, regulates the cemeteries, funeral parlours and crematoria and promotes safe and healthy environments. In the CCT's case, it also managed a portion of the local clinics in the metro during the pandemic.

The CCT is considered the best-run metro in the country, achieving consistently good audit opinions [36]. It has seen steady population growth (albeit at a slowing annual growth rate) from almost 3.1 million people in 2002 to 4.6 million in 2020, putting pressure on limited resources [5]. Almost 18% of households live in informal dwellings and 12% depend on community standpipes for clean water access [37], which hindered social distancing and good hygiene, which are essential for reducing the spread of COVID-19.

## 3. Methods

The CCT's COVID-19 Crisis Coordinating Team won the international Apolitical Global Public Service Team of the Year award for COVID-19 Rapid Responders in 2020 [38]. There are thus lessons for other city governments to learn, particularly on the data-driven crisis response in both the global North and global South and this was the main motivation for a CCT case study to be documented, analysed and published in the public domain. South Africa is a good country for this type of case study as it has developing country challenges and characteristics but enough good quality data and capabilities for a data-led response. This Cape Town case study is limited to the data component of the CCT's COVID-19 response due to the breadth and depth of the work, and the fact that other components were being documented elsewhere.

The research involved an evaluation and synthesis of documentation and personal experience relating to the COVID-19 data work stream in the CCT. Three main sources of information were used. Firstly, research permission was obtained from the CCT to access non-public documents and understand and discuss the development of the COVID-19 response. Secondly, two authors were intimately involved in leading, designing and working on the CCT COVID-19 data work stream through their roles at the Policy and Strategy Department in the CCT. Their personal notes, alongside select authorised documents and presentations, were reviewed and checked against the information gathered from other sources. Third, a structured interview questionnaire was designed as input for a close-out report on the work carried out by the CCT COVID-19 data work stream from March to July 2020. This included two parts: questions on need, process, stakeholder engagement, products, timelines and lessons for each main deliverable, and questions on general lessons and reflections on the data work stream, the CCT COVID-19 response, the CCT recovery and the CCT Data Strategy and its implementation. Semi-structured interviews were then held with most of the CCT staff within the data work stream who were directly involved on a daily basis in developing and operationalising the COVID-19 response and also with external consultants who assisted with the main data-driven products. Each interview was transcribed and assessed to understand the process, reflections and lessons learned. The findings were presented to the COVID-19 data work stream team and executive director for additional input and discussion before the close-out report was finalised. This close-out report was used as an input for this paper.

## 4. The City of Cape Town's Data-Driven Response

*4.1. Overview of the CCT COVID-19 Response*

The CCT adopted a dynamic operational framework for its COVID-19 response. This was based on two core insights that the city would cycle in and out of waves of the pandemic, and that the government would have to prioritise critical risks to protect the population due to limited resources and time. The CCT identified four critical systems—the

health system, disaster system, essential city operations and corporate city operations. The health system referred to the care of patients (COVID-19 and normal burden of disease). The disaster system referred to the combined safety, security and emergency response services of the city that need to coordinate in times of disaster. Essential city operations referred to the basic services of the city, e.g., water and sanitation. Corporate city operations covered all institutional systems and services that need to be sustained for the government to function, e.g., HR and finance. The framework recognised that as the city moved in and out of waves, these different systems would experience different levels of stress at different times. The city needed to understand how the pandemic was evolving locally and have data and insight into how the critical systems were responding. In addition to the four systems, there was a dedicated approach to informal settlements that recognised their vulnerability.

Taking the lessons from the CCT's disaster response planning to the 1:590 year drought in 2017/2018 on board [10], the CCT established the COVID-19 Coordination Committee (CCC). The committee ensured that the CCT was able to respond to the effects of COVID-19 in a timely manner. A dual response was decided on, with the City Manager and Mayor in charge of oversight (i.e., making and approving interventions) and the CCC preparing and implementing the needed programmes. Establishing the CCC meant the CCT had a single executive director (seconded full-time) leading the response, working across the traditional silos of the CCT. This allowed data to be used across the silos and a consistent data-driven decision approach to prevail. The CCC created work streams (light blue boxes in Figure 2) that formed a critical part of the execution structure and constituted the strategic management component of the COVID-19 response. These work streams, consisting of project teams across the CCT, provided the CCC with the tools for understanding information flows, risks and trade-offs across the broader planning efforts. At the point of conceptualising this unified response structure, it was recognised that the pandemic was unlikely to be a single wave and that the city would be cycling in and out of the crisis. It would therefore be critical for the response to be data led, guided by epidemiological expertise.

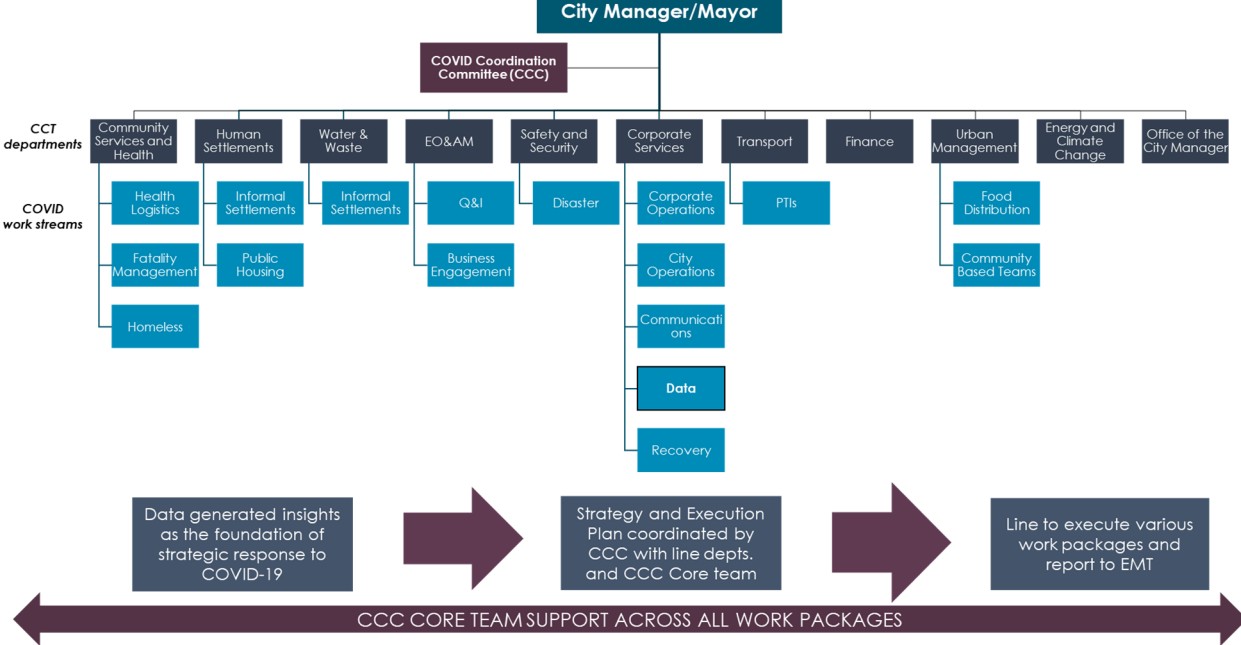

**Figure 2.** Overview of the City of Cape Town (CCT) COVID-19 response. Reprinted/adapted with permission from [39]. 2020, City of Cape Town. The City Manager and Mayor oversaw the COVID-19 Coordination Committee (CCC), which was responsible for the COVID-19 response and establishing and managing the needed work streams (light blue boxes). (EO and AM refer to Economic Opportunities and Asset Management; Q and I to Quarantine and Isolation, and PTIs to Public Transport Interchanges).

One of the key work streams established was the data work stream, with the initial aim of understanding the pandemic's impact on the CCT's ability to perform its service functions but quickly expanded to analysis that determined the risk level, pace and focus for each area of response. This entailed collecting data, managing data-sharing agreements, analysing patterns, unpacking the epidemiological models, and developing insights and projections that were communicated to inform strategies and other management tools such as logistics planning and fatalities management. Established in March 2020, this multi-disciplinary team comprised nine employees from different CCT departments and with diverse skill sets, all contributing to developing valuable insights from data. Notably, the CCT's epidemiologist formed part of the data work stream, with the data work being guided by her expertise. The rapid deployment of the team was enabled by the fact that the CCT was investing in data systems and capabilities for 20 years and accelerated this in 2018 (prompted by the drought) with the creation of a CCT Data Strategy and a data science environment, facilitating the deployment of open source software and analytics.

The data work stream underpinned all four phases of the CCT response, from planning and preparation to recovery (Figure 3). It produced nine main deliverables, most of which are described in the following subsections.

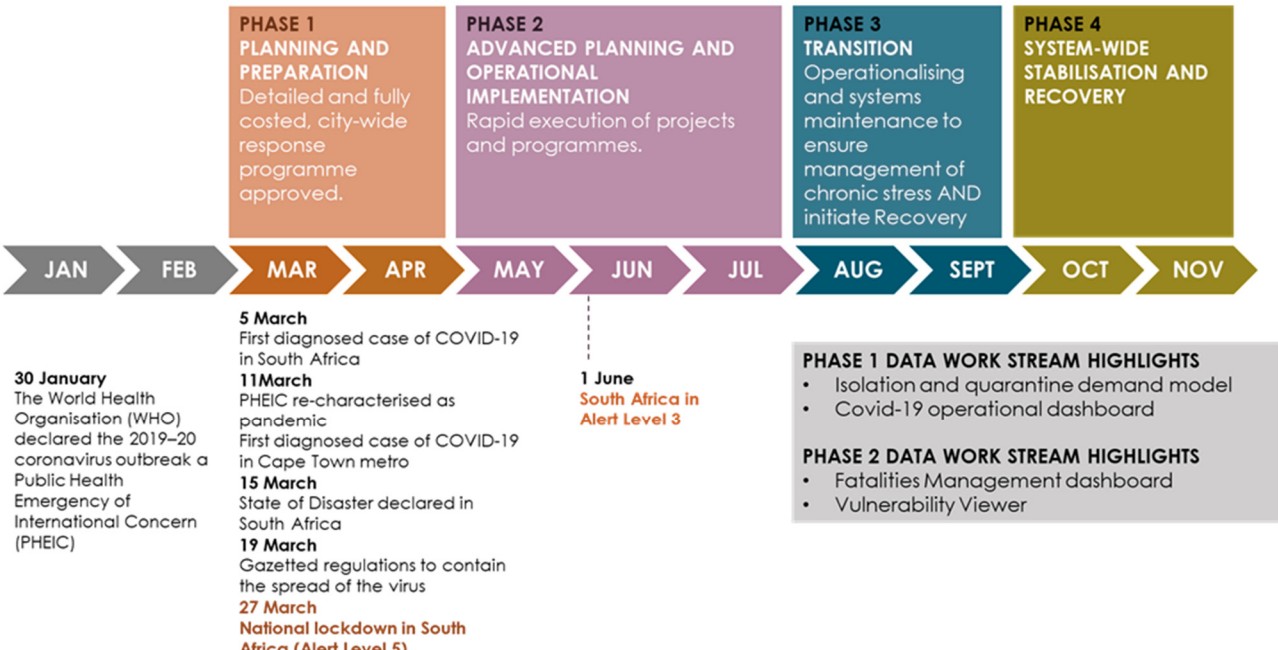

**Figure 3.** City of Cape Town (CCT) COVID-19 response timeline and phases throughout the pandemic. Reprinted/adapted with permission from [39]. 2020, City of Cape Town.

### 4.2. Epidemiological Model

Due to the multifaceted nature of the pandemic, the CCT decided early on to utilise an external epidemiological (epi-) model. The two main reasons were (1) it was too complex and high risk to try to build their own and (2) the city needed something to provide guidance on what to expect in terms of the scale of infections and deaths. Although the national response was being informed by the National Institute of Communicable Diseases' (NICD's) modelling, it was initially not available to other spheres of government. As such, the Western Cape Government's health department had an epi-model developed with academic partners, whilst also using its pre-existing relationships (with NICD and local private laboratories) to obtain COVID-19 test data. The epi-models used in the CCT initially originated from the provincial health department and were later updated to the national

model, both with minor local adjustments, such as city population and local COVID-19 case data. The epi-model was used as an input for various deliverables, such as Q and I, fatalities and clinic capacity estimates. The team tracked the course of the pandemic against the models to develop strategic insights to inform decisions about the pandemic's trajectory.

*4.3. Quarantine and Isolation Planning and Costing*

At the start of the pandemic (March 2020), the South African national government announced that public Quarantine and Isolation (Q and I) facilities be used for those unable to do so safely at home, and requested assistance from provincial and local government to urgently plan and prepare for public Q and I sites. Provincial governments were required to identify facilities, and develop appropriate plans for facility capacity and resources, including sufficient PPE per site, as well as transportation to and from Q and I sites. The CCT immediately recognised the need to rapidly plan and execute for public Q and I facilities ahead of the anticipated first wave and assisted the Western Cape Provincial Government. Moreover, the CCT also recognised the need, and took the initiative, to identify overall financial costs based on these estimates.

The epi-model, alongside information provided by the CCT's epidemiologist and province health experts (including assumptions on the average number of contacts per case or the percentage of the public dependent on public health care), was used to model the demand for Q and I associated planning requirements (e.g., PPE). Local Q and I estimates were validated against those from the provincial government, which estimated figures for the entire province. The results compared favourably, with enhancements made to both provincial and local government estimations due to collaboration.

In parallel to predicting total Q and I demand, the CCT developed a targeted Q and I planning parameter to give an indication of the number of vulnerable people who were unable to stay in their own homes and would most likely require public isolation and quarantine facilities. To identify these vulnerable people, a targeted Q and I Social Vulnerability Index (SVI) was developed. The targeted Q and I SVI combined variables from the 2011 Census such as disability, informal dwellings, population, household density and average household size, to identify these vulnerable households. This was a similar approach taken to the targeted SVI created for the 2018 drought [10] and the Socio-Economic Index that the CCT still uses for planning purposes. Yet, although the SVI was used for detailed "Day Zero" drought disaster planning, few of these indices had previously been incorporated into larger models. Here, the Q and I SVI was used as an input to the Q and I demand model to assist in determining the number of people that would need to be isolated or quarantined in public facilities over time, and thus the costing and logistic models. Two scenarios (low and high scenarios) were developed for the isolation and quarantine facilities (Figure 4). The CCT's Q and I SVI was incorporated into the Western Cape's own Q and I planning models. SVI maps also guided the facilities team to the spatial distribution requirements of the Q and I facilities in relation to population.

The estimated demand and subsequent financing projections from this work were the first in the country and were sent to the National Treasury for consideration and used to assess the feasibility of the proposed response. The analysis showed that the cost of meeting the Q and I commitment made by the national government would be financially unviable and was not pursued. Critically, this rapid analysis allowed that decision to be made in a timely manner and meant that resources and budgets could be focused elsewhere.

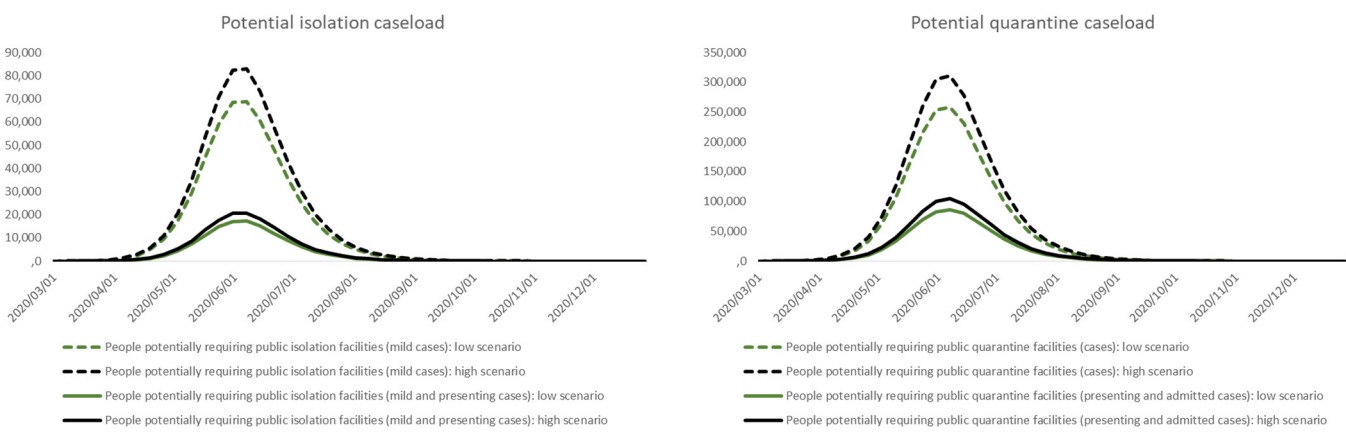

*Q&I demand forecasting forming the basis of costing and logistics modelling*

**Figure 4.** Quarantine and Isolation demand forecasting (using the Quarantine and Isolation Social Vulnerability Index (Q and I SVI)) showing the potential number of vulnerable people needing public Q and I facilities for a low and high scenario as of 7 April 2020. Reprinted/adapted with permission from [39]. 2020, City of Cape Town.

*4.4. Fatality Management*

Throughout the pandemic, the CCT continually monitored its progression and management against the epi-model, the country and internationally. The local epi-model showed instances of large volumes of deaths expected for the peak of the first wave in the winter of 2020. This was especially relevant to the CCT as local government is responsible for fatalities management and regulates cemeteries, funeral parlours and crematoria.

Although the CCT has limited room to assist private operators (i.e., undertakers or funeral homes), there were varying views in government on how to respond, with some arguing for the state to take over from private undertakers and run mass fatalities centres. Nevertheless, it was critical to ensure that deathcare services were not overwhelmed and the industry did not fail.

Before deciding on a response, the City worked with partners to map and analyse the value chain and determine the weak points based on a logistics analysis. Through this survey and analysis, private undertakers were identified as a critical link between where people die (at home or in a hospital) and being buried or cremated in a public or private facility. It was clear that private undertakers were best equipped to manage the load, but lacked a complete view of the total capacity across the system, which would assist them in managing and planning services.

The CCT identified that the most effective role it could play was to collect data on the overall system, in order to be aware of emerging problems and provide information to improve the coordination of the private actors, alongside having body storage and transport capacity on standby in case of a failure in the private provider system. The CCT could thus dramatically improve independent operators' capacity to make optimal decisions by giving them high-quality, real-time information on the capacity utilisation of other operators in the system. This could reduce the risk of complete or partial system failure. The elements analysed in the deathcare logistics model were captured in a summary dashboard shown in Figure 5.

To monitor the deathcare chain of services the CCT data work stream established a survey office to conduct daily (later reduced to weekly) surveys of capacity utilisation. This included PPE stocks, plant condition for mortuaries, cemeteries, crematoria and the CCT-run cemetery booking offices that schedule burials and cremations. Part of the CCT's call center was repurposed to facilitate the surveys in response to COVID-19 waves as needed. The call centres were deactivated in the troughs and reactivated as a wave re-emerged (see

Figure 1). An initial daily status report, using the survey, the epidemiology model outputs, the National Department of Home Affairs' capacity to issue death certificates and data on the pandemic's progression was created. The daily reports informed all deathcare service participants (that were opted in) of stress points across the chain of services. From this combined data, an internal fatalities dashboard was created to assist with CCT planning decisions including rearranging more funerals on weekdays (rather than weekends).

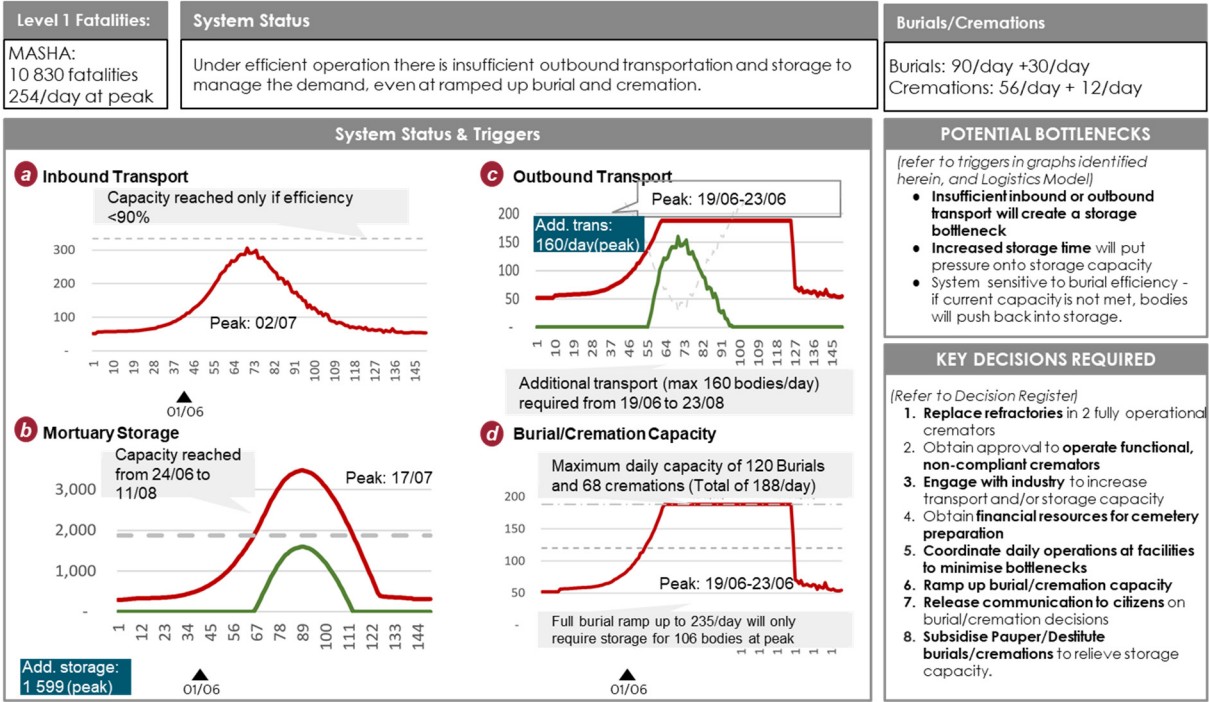

**Figure 5.** City of Cape Town's deathcare services capacity assessment and logistics model where the red line denotes the current scenario demand and the green line denotes the additional capacity required. Graphs x-axes show days while y-axes show number of bodies transported/stored or number of burials/cremations. Reprinted with permission from [40]. 2020, City of Cape Town.

The dashboard, created using R in the CCT's data science environment, contained sensitive data, combining the survey and critical internal data (e.g., the cemetery booking data) with external data (e.g., COVID-19 fatalities and hospitalisations) providing a holistic overview of the deathcare industry. Specifically, it consisted of multiple pages highlighting critical information for fatalities management, such as fatalities forecasts based on deaths and COVID-19 hospitalisations (Figure 6), monitoring the current situation against select triggers points, spatial data with maps of the various resource capacity levels across the industry, time series charts and stress plots which captured the deathcare industry's perception of their capacity.

The CCT also developed mini-dashboards containing selected data available to deathcare operators. In the normal course of death care in Cape Town, an individual operator has very little knowledge of the capacity of the entire system. For example, through informal contacts, undertakers may know the circumstances of the operators immediately upstream (e.g., transporters and hospitals) and downstream (local cemetery or crematoria booking office schedules), but specific data is not shared. Even though the CCT had limited room to assist private operators, the CCT data workstream with support from the call centre agents and internal data pipelines collated close-to-real-time data on the capacity utilisation and developments in the system from the hospital to the cremation. This was possible through existing data-sharing agreements and collecting data from the participating undertakers. The data collection process was activated on demand just before predicted waves, alongside the mini dashboards.

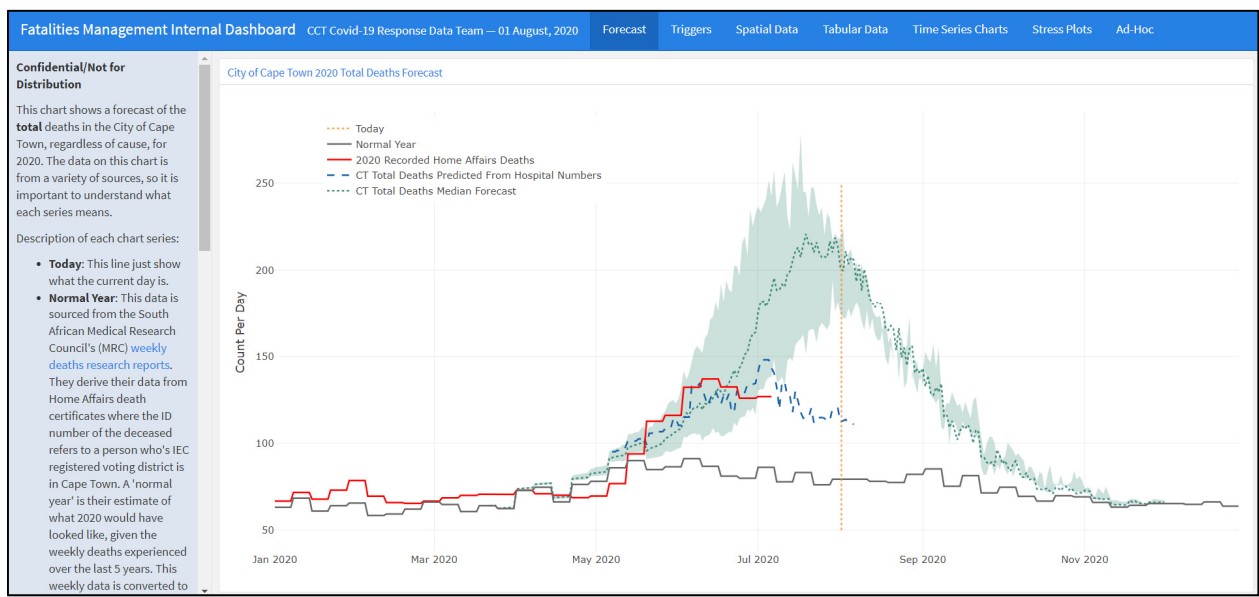

**Figure 6.** City of Cape Town's internal fatalities dashboard showing 2020 total deaths compared to a normal year and forecasted excess COVID-19 deaths. Reprinted/adapted with permission from [39]. 2020, City of Cape Town.

*4.5. Operations Dashboard for Executive Management*

Within the CCT, data are often both internally and externally communicated through static reports, such as the 2011 Census statistics for Cape Town (e.g., [41]). However, in response to the evolving spread of the virus, senior leadership and operational teams required real-time data to better understand potential impacts. Specifically, CCT officials required real-time data on the progression of COVID-19 nationally, provincially and locally in the context of (1) the virus transmission and (2) among vulnerable people, in an easy-to-understand format.

Dashboards were occasionally used within the CCT. However, they take time to design (working closely with end-users) and are built in traditional commercial platforms (e.g., Environmental Systems Research Institute (ESRI)). These platforms were unsuitable to collate and communicate the pandemic data due to the quick turnaround time required. As a result, the CCT data science team (as part of the data work stream) developed dashboards in Python and R to display COVID-19 and COVID-19-related data. This included live COVID-19 cases, hospitalisations, ICU admissions, recoveries and deaths but also comparative tracking to the epi-model (Figure 7). The main COVID-19 operational dashboard compared the progression of the virus to the rest of the world alongside scenario outcomes, and benchmarked Cape Town against other countries, provinces and cities, and was presented to internal leadership for real-time decision-making. Data was sourced from internal and external parties, such as the provincial Department of Health and CCT line departments, and data-sharing agreements were put in place with external parties.

Notably, the operational dashboard also included the geographical staff resourcing and capacity across the CCT, staff COVID-19 infections, CCT COVID-19 expenditure and CCT service requests and backlogs. For the latter, the public can log service requests capturing any challenges they might be facing with service delivery, e.g., water outages. These data are then collated to understand any service delivery challenges spatially and to measure the backlogs in addressing these requests. One of the big concerns in the lead-up to the first wave was concern about how COVID-19 would impact critical service delivery staff, and therefore essential services. If sufficient numbers of staff from a service were impacted, there could be a service failure leading to new risks (e.g., water and sanitation-related disease outbreaks).

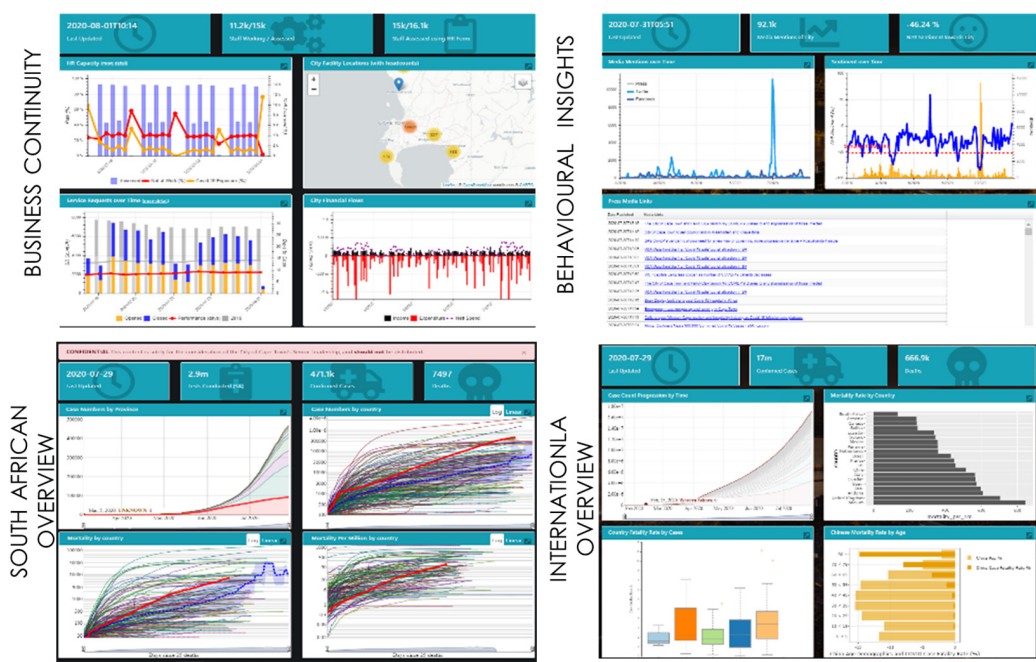

**Figure 7.** The City of Cape Town's COVID-19 operational dashboard as created by the data work stream showing business continuity (top left), behavioural insights (top right), South African overview (bottom left) and international overview (bottom right). Reprinted/adapted with permission from [39]. 2020, City of Cape Town.

*4.6. Vulnerability Viewer*

To support the national shift in health strategy in May 2020 towards "hotspots" and vulnerability [42], the data work stream developed the Vulnerability Viewer to assist operational CCT COVID-19 response teams (Figure 8). As areas of vulnerability differed depending on the particular response measure and evolution of the disease, it was essential to build a dynamic tool to provide relevant and insightful vulnerability profiles to assist operational teams in proactively targeting interventions in vulnerable places and spaces in the city.

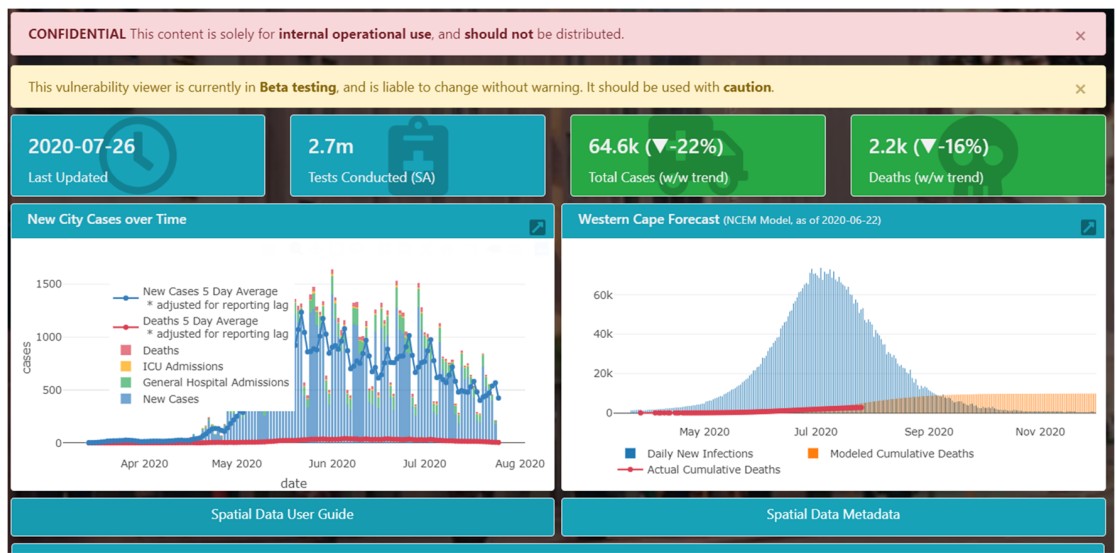

**Figure 8.** City of Cape Town's COVID-19 Vulnerability Viewer dashboard showing the pandemic progression across the City and the Western Cape province. Reprinted/adapted with permission from [39]. 2020, City of Cape Town.

The viewer visualised cases across the health districts and 500-hectare polygons /geographical areas with relevant contextual information such as population, old-age homes, areas of informality, shopping centres, South African Social Security Agency (SASSA) pay points, and a provincial socio-economic vulnerability index (SEVI) (Figure 9). The Vulnerability Viewer was developed quickly, becoming operational in June 2020, and was rolled out within the CCT soon after with continual improvements. This was possible because of the cross-functional team and the substantial existing work and investment the team could leverage, such as the existing sub-metro spatial population estimates, socio-economic maps, dashboard scripts and the data science environment and data science capabilities which enabled the use of open source tools. Key stakeholders including operational staff from health, human settlements and communications were consulted during development. A user guide was created, with the team offering support through weekly presentations and training "champions" to showcase the usability and power of the tool in critical operational meetings.

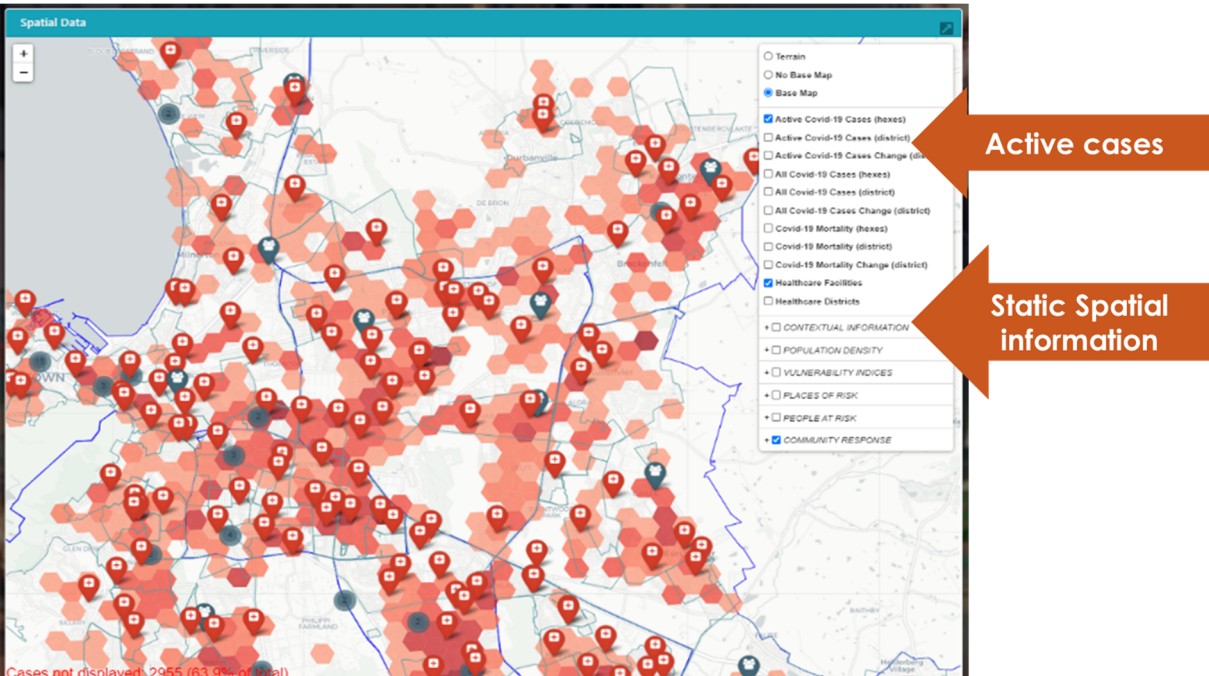

**Figure 9.** COVID-19 Vulnerability Viewer dashboard spatial analysis layer showing current active COVID-19 cases and health care facilities in the City of Cape Town. Reprinted/adapted with permission from [39]. 2020, City of Cape Town.

The commonality of the platform improved the coherency of CCT's COVID-19 resource response, at the correct times. For example, the Communications department used the Vulnerability Viewer to prioritise its COVID-19 communication campaigns targeting COVID-19 hotspots, big shopping centres and SASSA pay points. Similarly, the Western Cape health department developed the vaccination campaign with data from CCT viewers/dashboards, targeting SASSA pay points.

Prior to the use of live dashboards, presentations were the primary mode of communicating data to senior leadership. The data work stream hoped that there would be less reliance on presentations due to the replication of results (from the developed tools) within them. Whilst the tools were well received across the CCT, leadership and operational decision makers still preferred results to be presented, directly engaging with those who procured the data and developed the tools. Select individuals were identified to champion the Vulnerability Viewer in their respective forums, which proved to be successful. In the future, training operational champions to present content to senior leadership (as opposed

to the dashboard creators and data analysts) will be explored. These operational champions will have complete oversight of the models, data, logistical challenges and constraints enabling achievable and actionable recommendations.

### 4.7. Data sharing Agreements

Features of COVID-19-specific dashboards and their effectiveness are well documented [43–45], but the specific internal data procurement process for generating these tools is often unclear. Data used throughout the aforementioned CCT dashboards, models and/or tools originated from open, internal and external (to the CCT) sources. However, internal administrative and data system maturity barriers (such as data extraction from the CCT's administration software, Systems Application and Products (SAP)) frequently precluded rapid data integration. Similarly, external COVID-19 case data and the epidemiological outputs were owned by the provincial/national government, whilst the South African Medical Research Council (SAMRC) reported death data sourced from the Department of Home Affairs (with a two-week time delay). As prior data agreements were not in place, even for other levels of the South African government, facilitating those agreements under time pressure presented a significant challenge in being able to use data to inform decision-making.

To gather these datasets for use within the CCT's dashboards and tools, relationships with the relevant responsible departments had to be established. In each case, significant time was spent by the CCT's data work steam and legal team in writing and reviewing individual Memoranda of Understanding (MoU) along with creating data sharing pipelines and detailing data access. The CCT attempted to standardise MoUs to ensure continual access to data and, once in place, the CCT accessed near real-time COVID-19 data and epidemiological models. These data and models were implemented as previously described.

During the data procurement process, it became evident that some existing systems were not fit to communicate data in a meaningful way. Combined with poor data coordination and differing levels of data maturity across the CCT, data was inefficiently shared, which delayed informed decision-making. Once common and achievable objectives were established and the value of data-driven approaches demonstrated, such as assisting operation teams with their individual pandemic responses across the entire CCT, data sharing improved. Nevertheless, despite the agreements reached, several MoUs were quite restrictive with data not being used to its full potential. The CCT would greatly benefit from developing long-lasting MoUs (for external partners/datasets), and establishing clear data owners for various key datasets (e.g., embedded data capacity), alongside an agreed CCT-wide data usability standard.

## 5. Discussion

### 5.1. The Importance of Data

The COVID-19 pandemic presented a unique set of challenges for the CCT and required completely new work streams and operating procedures. The response and tools (as detailed above) aimed to address the ever-changing challenges brought on by the pandemic. The CCT's experience with a 1:590-year drought in 2017/2018 [10] accelerated the creation of a CCT Data Strategy and a data science environment, facilitating the deployment of open-source software and analytics. Having a data-driven response was more challenging in the COVID-19 crisis, however, due to the rapid nature of the pandemic, the scientific understanding was constantly evolving and key data systems/sources are split between different spheres of government (national, provincial and city). Table 1 summarises the four key challenges and the CCT's COVID-19 data-driven response to each. It shows that data was led strategically as well as operationally.

**Table 1.** Summary of the challenges faced by the CCT along with the developed tools, the data inputs and the decisions they informed.

| Challenge | Tool | Data Inputs | Example of Decisions Informed |
|---|---|---|---|
| Quantifying costs for COVID-19 initiatives | • Q and I Social Vulnerability Index (SVI) <br> • Q and I demand model | • Provincial government epidemiological model and health data <br> • Census data | Assisted in understanding the potential demand of public Q and I facilities, identifying facilities and the cost thereof, e.g., the analysis showed that the comprehensive public Q and I care would be financially infeasible. |
| Dealing with a surge in fatalities | • Internal fatalities management dashboard <br> • External mini dashboards for private operators | • Internal fatalities management data <br> • Private operator data <br> • Provincial health data <br> • South African Medical Research Council reported death data <br> • Home Affairs death data | Assisted with capacity decisions, e.g., when to push to schedule more funerals on weekdays (rather than weekends) to spread the load and not overwhelm deathcare services. |
| Guiding scarce public resources to respond to an evolving crisis | Three dashboards: <br> • COVID-19 operational dashboard <br> • Fatalities management dashboard <br> • Vulnerability Viewer | • Multiple internal static spatial datasets such as clinics, SASSA pay points, etc. <br> • Provincial health data <br> • City service delivery capacity <br> • External datasets | Assisted in prioritising and creating targeted interventions, e.g., the Communications department used the Vulnerability Viewer to prioritise their COVID-19 communication campaigns. |
| Data sharing | • Relationships with internal and external data providers <br> • MoUs <br> • Data pipelines | • See data sources above | Assisted in realising the decisions above. |

The data-driven response detailed in Table 1 above, within national and local considerations, was achieved through a combination of a clear delivery structure through the CCC, a commitment to a response led by data and evidence and prior investment into data capabilities. Since 2018 the CCT data science unit has built a data science environment and data flows necessary to rapidly deploy open-source software to enable analytics. When the pandemic hit, data staff members were retasked to the COVID-19 data work stream and thus onto the pandemic response. Moreover, there has been a systematic investment over the past decade in a data-driven approach to corporate programme management, with the building up of a Corporate Portfolio, Programme and Project Management team (C3PM) and supporting data systems. Combined, these strategic approaches and investments in data systems enabled data to play an essential role in guiding senior leadership to the most appropriate decisions during the pandemic.

Prior to this initiative, the CCT collected data to inform certain policies and strategies. However, the speed of analysis has never had such importance in avoiding complete system failure. Even during the 2018 drought, data was more readily available and mostly within the CCT's control. The fast-paced evolution of the virus required the CCT to respond appropriately by collecting and analysing large volumes of data from multiple sources to inform decisions. The spreadsheets used (e.g., for track and trace) rapidly hit their limitations. Using open-source tools and data science skills allowed the CCT to automate data flows and derive meaning from a much larger volume of information in a more agile

way. To maximise efficiency, the CCT acted swiftly to overcome immediate technical, managerial and resource challenges. Through the new dashboards guiding CCT policy and facilitating data sharing between deathcare operators, the CCT was able to ensure effective resourcing and capacity management of the industry.

As noted within the introduction, a wide body of literature exists on both data-driven disaster responses and data-driven urban planning, but the immediate response and decision processes of city governments are poorly documented, especially in the global South. Similarly, whilst many studies demonstrate the effectiveness of data analysis in response to COVID-19 data, the majority of work is either at the national or sub-national levels in the global North with little to no informality. This paper has demonstrated and discussed that data-driven approaches in a developing city with high levels of informal housing, informal economic activity and multiple forms of deprivation, poverty and health issues (HIV, TB) can avoid system failure. South Africa has a lot of history of using data to combat HIV/AIDS and TB, but this is very much in the domain of the healthcare system. Capturing lessons about how a South African city government used data to guide a multi-sector response to a pandemic (not just health) is important for sharing lessons and identifying areas of future research.

### 5.2. Lessons Learned and Recommendations

From the CCT's experience during the COVID-19 and 2018 drought events, data played an integral role in guiding operational decision-makers. These processes stemmed from establishing foundational elements such as prior investment in flexible data systems, capabilities, data maturity and, crucially, partnerships across related internal and external government departments. Whilst time-consuming, any investments and data agreements must become an integral part of the business-as-usual arrangements. This means in the event of a crisis there are minimal barriers to sharing data for analysis and tool development, resulting in better-informed decisions, optimal resource usage, and in some circumstances, more lives saved. The deployment of tools using open-source software within this paper, such as the facilities management dashboard, Vulnerability Viewer and COVID-19 operational dashboard assisted with rapid responses that would not have been possible using proprietary software and traditional application development timelines or approaches. CCT's COVID-19 experience is informing the revision of its Data Strategy, including data analytics standards, open-source tools and the use of APIs to bring external data into the city.

However, data, analysis and subsequent visualisation are not solely sufficient to solve a crisis or disaster given the complex and social and economic challenges of cities, especially in the global South. A clear structure, with a programmatic response, that utilises all expertise is essential. Multi-disciplinary teams must be jointly composed of data engineers, data analysts, subject matter experts, project management specialists and strategy advisors to reach the most appropriate actions in supporting and aiding residents following shock events. Moreover, rapid assessments of crisis initiatives and feedback loops between different spheres of government are critical and should be embraced, as seen in the Q and I assessment.

Given that environmental shocks and stresses will likely increase in frequency [46], academic researchers must use these findings and reflections to understand the constraints in which city officials operate to effectively devise impactful and appropriate research. Similarly, metropolitan, regional and national governments must not delay in preparing cross-governmental data strategies, supported by programme design and implementation capacity, for relevant future risks in order to effectively serve the population through efficient and data-evidenced responses. City governments need to be prepared to implement new leadership and delivery structures, with data capability at their core, to respond to metro-scale shocks.

**Author Contributions:** All authors assisted in conceiving and/or designing the manuscript led by E.N.; E.N. outlined the challenges and data responses of the City of Cape Town and developed

the manuscript; A.M. and O.B. developed and edited the manuscript; H.C. initiated the paper and provided initial concept, oversight and manuscript edits; M.C. edited the manuscript. All authors have read and agreed to the published version of the manuscript.

**Funding:** This research received no external funding.

**Institutional Review Board Statement:** Research permission was obtained by E.N. and H.C. from the CCT to access non-public documents and understand and discuss the development of the COVID-19 response.

**Informed Consent Statement:** Not applicable.

**Data Availability Statement:** Data used in Figure 1 were sourced from the COVID-19 Data Hub: https://covid19datahub.io/ (accessed on 23 October 2022), [35]. The reproducible code for Figure 1 can be found on GitHub: https://github.com/andrewmaclachlan/CCT (accessed on 23 October 2022).

**Acknowledgments:** We acknowledge the following City of Cape Town (CCT) staff who led the development of the teams and tools outlined within this paper. (1) Hugh Cole, being the appointed data work steam lead. (2) Elmarie Nel and Natacha Berkowitz, leading the Quarantine and Isolation forecasting (Figure 4). (3) Riaz Arbi, leading the CCT internal fatalities dashboard with support from Moeneeb Abass (Figure 6). (4) Carol Wright, leading the business continuity information for the operational dashboard (Figure 7). (5) Gordon Inggs, Elmarie Nel and Annelise de Bruin, leading the vulnerability viewer dashboards (Figures 8 and 9). (6) Katherine Hyman for assisting with relevant contextual slides (Figure 3). (7) Ben Peters and the C3PM team for the logistics modelling. (8) Craig Kesson for overall leadership of the response.

**Conflicts of Interest:** The authors declare no conflict of interest. Please note that the views/opinions expressed in this publication are those of the authors.

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
