# Peer review of "Data-Driven Decision Making in Response to the COVID-19 Pandemic: A City of Cape Town Case Study"

_sustainability, doi:10.3390/su15031853_

Round 1

Reviewer 1 Report

The paper outlines specific challenges the city faced and its data-informed response in the areas of:

(1) quantifying costs for COVID-19 initiatives

(2) dealing with a surge in fatalities

(3) guiding scarce public resources to respond to an evolving crisis

(4) data sharing.

The paper demonstrate the real-term value in incorporating data into the decision making process and conclude by outlining key factors that cities and researchers must consider as a part of usual business to effectively assist their populations during times of stress and crisis.

Introduction, methods, results, discussion and conclusion are clearly presented.

Author Response

The authors thank this reviewer for their comments and it is evident that the points made in the paper have been interpreted correctly.

Reviewer 2 Report

Please see my comments attached.

Author Response

  1. This study takes on an important topic and provides a substantial amount of information that can be useful for COVID-19 or similar pandemic responses elsewhere. However, a flaw in the study, at least the way it is written, is a general lack of organization of the article as one based on the case study methodology. The authors may refer to one of the book chapters (or books) on case study methodology, including one by the pioneer of case study methodology, Robert Yin,--(e.g. Yin, Robert K. "Case study methods." (2012)).
  2. In its current form, the article reads like a review article with some figures included from grey literature without properly declaring the sources of data for those figures. The methods section is missing altogether. The authors must include a methods section right after the introduction section, and cover the following:
  3. The authors must describe their study design and what is the justification for calling it a case study
  4. Explain how they selected case study materials for their analysis, including whether primary or secondary data are used.
  5. What type of data did they analyze (qualitative, quantitative, mixed); and how were they analyzed, using which software? How did these data and variables correspond to their implied/stated study objectives of quantifying costs for COVID-19 initiatives, determining the COVID-19 surges in fatalities, and other quantifiable objectives? How was the thematic analysis of any qualitative data conducted? How were these qualitative data coded (using which software) and synthesized?

The authors thank the reviewer for their comments. We have reorganised the paper, including a Methods section, a Results section and a Discussion section. We have included references for the figures.

Here are some more specific points authors should address:

TITLE:

  1. Replace “;” with a “:” to indicate it is a subtitle. The use of the semi-colon is not a convention.

This has been changed.

ABSTRACT:

  1. The abstract needs to be rewritten. The authors have completely skipped the methods section. In addressing this comment, the authors should provide details in the abstract covering:
  2. Study design
  3. Data sources and details of which qualitative and quantitative data were included in the case study.
  4. A brief mention of qualitative and quantitative methods used to analyze the data included in the case study
  5. State the results clearly, listing the “key factors that cities and researchers must consider as a part of usual business to effectively assist their populations during times of stress and crisis.”

The abstract has been updated to include the methods and results.

INTRODUCTION:

  1. The authors have completely skipped the details about data-driven policymaking, including:
  2. Why an emphasis on data-driven policy making is essential in responding to COVID-19 pandemic?
  3. What are enabling factors for data-driven decision-making in general and policy decisions in particular?
  4. What is the state of information systems used in COVID-19 response in South Africa and elsewhere?

This is stated at the end of the introduction. We have added “to optimize limited resources” on line 93. It has also been included in the Discussion section.

  1. While the above details are skipped, the authors have provided some unnecessary details COVID-19 pandemic, without properly creating relevance of these details to their study.

We acknowledge this and have removed the unnecessary details.

  1. The entire first paragraph is not properly cited. Provide proper references to published literature to support the following statements in the first paragraph :
  2. “The 2019 Coronavirus (COVID-19) Pandemic….”
  3. “Although faced with the same virus,…”
  4. “In particular, countries with a high proportion…”
  5. “Cities have been particularly…

Citations have been added.

  1. Line 37-39: The statement should be properly cited.

We have cited it.

  1. Line 43-46: The statement should be properly cited.

We have cited it.

  1. Lines 59-65: The statement should be properly cited.

We have cited it.  

METHODS:

  1. After the “Introduction section”, the entire section titled “2. South African and Cape Town Context” seems out of place. Even though the research utilizes a Case study approach, the methods are expected right after the introduction section is concluded. See the other details states in the points 1 and 2.

We have kept this section as it provides important context to South Africa and Cape Town that would make the Introduction too wordy.

Reviewer 3 Report

Dear Authors,

The authors presented an approach of Data-driven policy making in response to the COVID-19 pandemic; a City of Cape Town case study.” The subject is interesting and within the scope of the journal. However, I think significant concerns are reasonable and could improve the manuscript’s quality before publication. Overall, I would recommend the publication of this manuscript subject to “major revision −taking into account precisely addressing the following comments in the revised version.

1.     Please rewrite the title in the following order: “Data-driven policy making in response to the COVID-19 pandemic: a case study of Cape Town City.”

2.     The abstract has flaws. Please make it precise with the following information: background, methods, results, and concluding remarks.

3.     The concluding remarks are absent in the abstract. I recommend stressing it in this section.

4.     I am not sure what is the main research hypothesis. What are the research gaps?

5.     What are the main contributions of this study?

6.     Justification for selecting input is required.

7.     For a research article, we generally follow the following structures: Introduction, Methods, Results, Discussion, and Conclusion. Which is currently absent in the present structures. For me, it is very difficult to maintain differentiation among various contents. Currently, the paper reduces the readability of the authors. Besides, few elements can be accommodated as supplementary documents because these unnecessarily increase the volume of the manuscript.

8.     The contents sometimes jump between topics without a clear direction in the discussion. I would have wished to see more information on the actual meaning of the findings and how the results add to the broader issue and the specific scientific field.

Author Response

  1. Please rewrite the title in the following order: “Data-driven policy making in response to the COVID-19 pandemic: a case study of Cape Town City.”

The authors thank the reviewer for their comments. This has been changed.

  1. The abstract has flaws. Please make it precise with the following information: background, methods, results, and concluding remarks.

The abstract has been updated to reflect the methods, results and concluding remarks.  

  1. The concluding remarks are absent in the abstract. I recommend stressing it in this section.

As above.

  1. I am not sure what is the main research hypothesis. What are the research gaps?

The gap in the research is lack of understanding and knowledge of the multidisciplinary processes used and innovative data tools developed in cities to respond to city-scale public challenges, like the COVID-19 pandemic.

The research gaps are noted within the abstract and throughout the paper. For example…

“we demonstrate the real-term value in incorporating data into the decision making process and conclude by outlining key factors that cities and researchers must consider as a part of usual business to effectively assist their populations during times of stress and crisis.”

  1. What are the main contributions of this study?

The contributions are listed at the end of the introduction, and in the discussion section.

“This paper will explore how the CCT’s data-driven developments assisted in responding to these overwhelming challenges to optimize limited resources. Previous literature has discussed the impact of COVID-19 on the economy [14], cities in a post-COVID world [15] and centralised government responses [16,17]. While some examples of da-ta-driven responses to public health crises have been published, such as New York City’s response to Legionnaires disease in 2015 [18], often the internal data advances and work streams within metropolitan governments are poorly documented despite being at the forefront of city decision making. In light of urbanisation and the need for urban resilience in developing countries, it is essential to understand the capabilities, constraints and process in which government data analysts and policy makers operate to design and implement crisis responses.”

  1. Justification for selecting input is required.

This has been done.

  1. For a research article, we generally follow the following structures: Introduction, Methods, Results, Discussion, and Conclusion. Which is currently absent in the present structures. For me, it is very difficult to maintain differentiation among various contents. Currently, the paper reduces the readability of the authors. Besides, few elements can be accommodated as supplementary documents because these unnecessarily increase the volume of the manuscript.

The paper has been reorganised to reflect these sections and improve readability.

  1. The contents sometimes jump between topics without a clear direction in the discussion. I would have wished to see more information on the actual meaning of the findings and how the results add to the broader issue and the specific scientific field.

This has been addressed by reorganising the paper’s findings into a Results section with subsections based on deliverables, and a Discussion section, which firstly explains the importance of data for decision making in city government, particularly in disaster situations, and secondly describes the lessons learned by the City of Cape Town government and recommendations for other city governments and academics to guide their future research.

Round 2

Reviewer 2 Report

The authors have adequately addressed all of my comments.  One minor need for improvement is in the methods. Although the authors declare in their article's title that the research design is a "case study," they have not mentioned the case study design in their methods sections. That should be done.

Author Response

The authors thank this reviewer for their comments. A new paragraph has been added at the beginning of the Methods section to address this.

Reviewer 3 Report

The manuscript is ready to publish in its current form. 

Author Response

The authors thank this reviewer for their review.